# Peficitinib Inhibits the Chemotactic Activity of Monocytes via Proinflammatory Cytokine Production in Rheumatoid Arthritis Fibroblast-Like Synoviocytes

**DOI:** 10.3390/cells8060561

**Published:** 2019-06-09

**Authors:** Yuzo Ikari, Takeo Isozaki, Yumi Tsubokura, Tsuyoshi Kasama

**Affiliations:** Division of Rheumatology, Department of Medicine, Showa University School of Medicine, Tokyo 142-8555, Japan; y.ikari@live.jp (Y.I.); tsubokura@cnt.showa-u.ac.jp (Y.T.); tkasama@med.showa-u.ac.jp (T.K.)

**Keywords:** peficitinib, rheumatoid arthritis, fibroblast-like synoviocytes, monocyte chemotaxis

## Abstract

**Background:** This study was performed to examine the effects of the Janus kinase (JAK) inhibitor peficitinib on fibroblast-like synoviocytes (FLS) obtained from patients with rheumatoid arthritis (RA). **Methods:** To examine the expression of JAK1, JAK2, and JAK3 in RA synovial tissue (ST) and FLS, immunohistochemistry was performed. We investigated the effects of peficitinib on interleukin 6 and IL-6 receptor responses in RA FLS. Phosphorylation of STAT was determined by western blot. To examine the functional analysis of peficitinib, we performed a proliferation and chemotaxis assays with FLS using THP-1 and peripheral blood mononuclear cells (PBMC). The inflammatory mediator expression of FLS was estimated by enzyme-linked immunosorbent assay. **Results:** JAK1, JAK2, and JAK3 were expressed in RA STs and FLS. Phosphorylation of STAT1, STAT3, and STAT5 in RA FLS was suppressed by peficitinib in a concentration-dependent manner. Peficitinib-treated RA FLS-conditioned medium reduced THP-1 and PBMC migration (*p* < 0.05) and proliferation of RA FLS (*p* < 0.05). Peficitinib suppressed the secretion of MCP-1/CCL2 in the RA FLS supernatant (*p* < 0.05). **Conclusion:** Peficitinib suppressed the JAK-STAT pathway in RA FLS and also suppressed monocyte chemotaxis and proliferation of FLS through inhibition of inflammatory cytokines.

## 1. Introduction

Rheumatoid arthritis (RA) is a chronic systemic autoimmune disease characterized by inflammatory synovitis, cartilage and bone destruction, and several systemic features [1]. In the intra-articular synovial tissue of RA patients, inflammatory cell infiltration by T lymphocytes, B lymphocytes, plasma cells, macrophages, neutrophils, mast cells, natural killer cells, and dendritic cells is witnessed. These inflammatory cells are activated, and inflammatory cytokines, such as tumor necrosis factor α (TNFα), and interleukin 6 (IL-6) are released [2]. Those cytokines then proliferate and activate fibroblast-like synoviocytes (FLS), which then secrete IL-6 and proteolytic enzymes to form pannus in joints thereby destroying cartilage and bone [3].

Janus kinase-signal transducer and activator of transcription (JAK-STAT) pathway have been identified as an important signaling pathway of various cytokines in RA (e.g., IL 6) [4,5]. Janus kinase (JAK) is a cytoplasmic protein tyrosine kinase associated with various cytokine receptors [6]. When the cytokines bind to membrane receptors, JAK is activated and STAT is phosphorylated to form [homo-dimers and hetero-dimers]. Dimer-formed STATs migrate into the nucleus and activate transcription of proinflammatory genes [7]. JAK has four molecules, namely JAK1, JAK2, JAK3, and TYK2. Pharmacological inhibition of JAKs was shown to efficiently block the downstream events associated with type I/II cytokines. The molecules of the signaling pathways, such as the JAK family are thought to be promising targets for RA treatment [7,8].

Peficitinib is a novel orally bioavailable JAK inhibitor under development for the treatment of RA. Peficitinib inhibits JAK1, JAK2, JAK3, and Tyk2 enzyme activity having 50% inhibitory concentrations (IC50) values of 3.9, 5.0, 0.71, and 4.8 nmol/L, respectively, and has moderate selectivity for JAK3 inhibition [9]. Peficitinib has been shown good efficacy in clinical trials [9,10,11], but elucidation of its mechanism of action in RA involving the inflammatory process is still inadequate. In this study, we evaluated the effects of peficitinib in RA FLS.

## 2. Materials and Methods

### 2.1. Patients

RA ST samples were obtained from patients undergoing arthroplasty. One patient was treated with tocilizumab and methotrexate for 11 months and had 2 years disease duration. The other was treated with abatacept for 3 years and had 6 years disease duration. These treatments could have affected cytokines pathways. All specimens were collected after obtaining informed consent. The study protocol was approved by the Showa University Institutional Review Board.

### 2.2. Cell Culture

Fresh ST samples were minced and digested in tissue enzyme digestion solution. FLSs were maintained in RPMI-1640 medium supplemented with 10% fetal bovine serum (FBS). The cells were seeded in 6-well plates (BD Biosciences, Bedford, MA, USA) at a density of 2 × 10^5^ cells/well and were maintained in a complete medium. After overnight serum starvation, the cells were stimulated with IL-6 (100 ng/mL; R&D Systems, Minneapolis, MN, USA) and IL-6 receptor (IL-6R; 100 ng/mL; R&D Systems).

### 2.3. Immunohistochemical Analysis

The frozen ST samples obtained from RA patients were stained by the immunoperoxidase method. The slides were fixed in cold acetone for 20 min and were washed with phosphate buffered saline (PBS). Following incubation with 3% H_2_O_2_ for 10 min to block endogenous peroxidase, the ST samples were blocked with 20% FBS and 5% goat serum in PBS for 1 h at 37 °C. JAK1, JAK2 (Cell Signaling Technology, Danvers, MA, USA), JAK3 antibodies (Abcam, Cambridge, MA, USA), and rabbit IgG were used for overnight at 4 °C. The ST samples were washed with PBS, and biotinylated goat anti-rabbit IgG (1:200 dilution in blocking buffer; Vector Laboratories, Burlingame, CA, USA) was added and incubated for an additional 1 h at 37 °C. After washing, antibody binding was detected with a Vectastain ABC standard kit (Vector Laboratories, Burlingame, CA, USA) and 3,3’-diaminobenzidine (DAB; Vector Laboratories, Burlingame, CA, USA) as the chromogen. Hematoxylin staining was performed and the images were captured at 200× magnification.

### 2.4. Western Blot Analysis

The cells were seeded in 6-well plates (BD Biosciences, Bedford, MA, USA) at a density of 1.2 × 10^5^ cells/well and were maintained in RPMI-1640 medium supplemented with 10% FBS. After overnight serum starvation, the cells were left unstimulated or were stimulated with IL-6 (100 ng/mL) and IL-6R (100 ng/mL) for 10 and 30 min. After adding peficitinib (MCE), the cells were stimulated with IL-6 (100 ng/mL) and IL-6R (100 ng/mL) for 10 min with or without adding peficitinib at different concentrations (0.1, 1, and 5 μM). RA FLSs were lysed with a lysis buffer (100 μL/well; CelLytic^TM^, Sigma-Aldrich, ST. Louis, MO, USA) containing protease inhibitors. The protein concentration of each extract was determined using a Pierce^TM^ BCA Protein Assay Kit (Pierce Biotechnology, Rockford, IL, USA). With Novex™ 10% Tris-Glycine Mini Gels WedgeWell™ format 10-well (Pierce Biotechnology, Rockford, IL, USA), electrophoresis was performed on the cell lysates after equal protein loading and the proteins were transferred onto Immuno-Blot^®^ PVDF Membrane For Protein Blotting (Bio-Rad, Hercules, CA, USA) using a traditional wet transfer apparatus (Bio-Rad). STAT1, STAT3, STAT5, phospho-STAT1, phospho-STAT3, and phospho-STAT5 antibodies (Cell Signaling Technology, Danvers, MA, USA) were used as primary antibodies after blocking with 5% skimmed milk. HRP-conjugated anti-rabbit IgG (Cell Signaling Technology) was used as the secondary antibody. ECL Prime Western Blotting Detection Reagent (GE Healthcare, Chicago, IL, USA) was used as a detection reagent. The immunoreactive protein bands were visualized by enhanced chemiluminescence (Amersham Biosciences, Piscataway, NJ, USA).

### 2.5. Proliferation Assays

RA FLSs were seeded at 1 × 10^4^ cells/well in 96 well plates and were maintained in RPMI-1640 medium supplemented with 10% FBS. After overnight serum starvation, the cells stimulated with IL-6 (100 ng/mL) and IL-6R (100 ng/mL) were divided into two groups: treated with peficitinib (5 μM) for 24 h, and untreated cells. FLS proliferation was determined using CyQUANT^®^ Cell Proliferation Assay Kit (Life Technologies, Carlsbad, CA, USA) according to the manufacturer’s instructions. For the assay, the cells were lysed and the total cellular nucleic acid was estimated using fluorescence at 520 nm emission after excitation at 480 nm.

### 2.6. Chemotaxis Assays

For THP-1 or peripheral blood mononuclear cells (PBMC) chemotaxis assays, a 48-well Boyden chamber with 5 μm polycarbonate membrane was used. The lower wells were filled with the stimulus solution. Synovial fluid (SF) [1:50 dilution (in 0.5% BSA/RPMI)] was used as the positive control, and 0.5% BSA/RPMI was used as the negative control. THP-1 cells or PBMC in 0.5% BSA/RPMI at 1.2 × 10^6^/mL were added to the upper wells and incubated at 37 °C for 90 min. After incubation for 90 min at 37 °C in an incubator, the membrane was stained with DiffQuick. The migrated cells were counted by a blind observer. Three high-power (400×) fields were counted in each well, and the results were expressed as the number of cells per high power fields (HPFs).

### 2.7. Enzyme-Linked Immunosorbent Assays (ELISA)

RA FLS supernatant was obtained from RA FLS-conditioned medium stimulated with IL-6 and IL-6R (R&D System) with or without adding peficitinib (5 μM). RANTES/CCL5, MCP-1/CCL2, MMP3, fractalkine/CX3CL1, ENA-78/CXCL5, and IL-8/CXCL8 in the cell supernatant was measured using an ELISA kit (R&D Systems, Minneapolis, MN, USA) following the manufacturer’s protocol. Briefly, 96-well plates were coated with mouse anti-human antibody as the primary antibody, and RA FLS supernatant was added. The plates were washed, and biotinylated goat anti-mouse antibody was added followed by the addition of streptavidin-horseradish peroxidase. The plates were developed using tetramethylbenzidine substrate (TMB, Sigma-Aldrich, ST. Louis, MO, USA) and a microplate reader. The absorbance was recorded at 450 nm.

### 2.8. Statistical Analyses

The variance of the data was evaluated using the F test. The data were analyzed using Student’s t-test for equal variance and Mann–Whitney U test for unequal variance. The data are reported as the mean ± standard error of the mean (SEM). A *p* value of <0.05 was considered statistically significant.

## 3. Results

### 3.1. Expression of JAK1, JAK2, and JAK3 in RA STs and FLSs

To determine whether JAK1, JAK2, and JAK3 were expressed in RA ST, immunohistochemistry was performed. We found that JAK1, JAK2, and JAK3 were expressed in RA ST (Figure 1A). JAK1 and JAK3 were observed in RA ST lining layers, indicating that the cells in the synovial sublining area expressed high levels of JAK1 and JAK3. JAK2 was expressed entirely within the RA ST cell nucleus. JAK3 staining was observed in the RA ST, indicating that the cells in the synovial lining cells and sublining area expressed high levels of JAK3. For further examining the expression of JAK1, JAK2, and JAK3 in RA FLSs, we examined the FLS isolated from RA ST. JAK1, JAK2, and JAK3 (Figure 1B). JAK2 expression was confirmed by nuclear staining, and we confirmed that JAK1, JAK2, and JAK3 were expressed in RA STs and FLS.

### 3.2. IL-6 and IL-6R Activated the JAK-STAT Pathway in RA FLS

To determine whether IL-6 and IL-6R activate JAK-STAT pathway in RA FLS, western blot was performed. Activation of JAK-STAT pathway was confirmed by augmenting the phosphorylation of STAT1, STAT3, and STAT5. Representative western blot images signified that the expression of phospho STAT1, phospho STAT3, and phospho STAT5 were significantly higher after 10 min of stimulation with IL-6 (100 ng/mL) and IL-6R (100 ng/mL) as compared to that without stimulation (Figure 2A−F). Total STAT5 had two bands (p-STAT5A and p-STAT5B). This result was considered to be the influence of the antibody preparation. We demonstrated that the stimulation of IL-6 and IL-6R could activate the JAK-STAT pathway in RA FLS.

### 3.3. Peficitinib Inhibited the JAK-STAT Pathway in RA FLS

To determine whether peficitinib regulates the JAK-STAT pathway in RA FLS, western blot analysis was performed. Suppression of the JAK-STAT pathway was confirmed by reduced phosphorylation of STAT1, STAT3, and STAT5. RA FLS were stimulated with IL-6 (100 ng/mL) and IL-6R (100 ng/mL) for 10 min after the RA FLS were treated with peficitinib (0.1, 1, and 5 μM) for 24 h. Phosphorylation of STAT1, STAT3, and STAT5 in the RA FLS was suppressed by peficitinib in a concentration-dependent manner (Figure 3A−F). We confirmed that peficitiib suppressed the activation of JAK-STAT pathway stimulated with IL-6 and IL-6R.

### 3.4. Peficitinib Inhibited the Monocyte Chemotactic Activity

Furthermore, the peficitinib treated RA FLS-conditioned medium reduced THP-1 migration as compared to the untreated RA FLS-conditioned medium (number of THP-1 cells migrated ± SEM; 42 ± 3 vs. 66 ± 6, *p* < 0.05; Figure 4A). The peficitinib-treated RA FLS-conditioned medium also reduced PBMC migration as compared to the untreated RA FLS-conditioned medium (number of PBMC migrated ± SEM; 36 ± 5 vs. 63 ± 9, *p* < 0.05; Figure 4B).

### 3.5. Peficitinib Inhibited the Proliferation of RA FLSs

We proved the existence of the JAK-STAT pathway in FLS by peficitinib. Further, we performed a proliferation assay with FLS for functional analysis. Peficitinib (5 μM) was added to FLS and was stimulated with IL-6 and IL-6R. Peficitinib-treated RA FLS showed a 14% reduction in proliferation as compared to the untreated RA FLS (Figure 5).

### 3.6. Peficitinib Suppressed the Secretion of Inflammatory Mediators in RA FLS

We found that peficitinib is involved in the suppression of FLS proliferation, and inhibits the chemotaxis of THP1 and PBMC. Finally, we found that peficitinib inhibited the secretion of inflammatory mediators in RA FLS. MCP-1/CCL2 in RA FLS supernatant was suppressed after adding peficitinib when compared to that without adding (160 ± 65 pg/mL vs. 846 ± 107 pg/mL, *p* < 0.05; Figure 6B). RANTES/CCL5, MMP-3, fractalkine/CX3CL1, ENA-78/CXCL5, and IL-8 in RA FLS supernatant were not significantly differences as compared to the untreated medium (Figure 6A,C–F). From these results, suppression of chemotaxis of THP-1 and PBMC was confirmed through inhibition of MCP-1/CCL2 in the RA FLS supernatant.

## 4. Discussion

Peficitinib is a JAK inhibitor that has been developed for the treatment of RA. Peficitinib has shown efficacy in clinical trials [9,10,11]. In a randomized, double-blind, placebo-controlled phase IIb study, the effectiveness and safety of a single dose of peficitinib in active RA patients was demonstrated [9]. However, its mechanism in RA in the presence of an inflammatory process is still not clear. To date, no study has been performed to examine the JAK-STAT pathway or the effects of peficitinib in RA FLS isolated from RA ST. This is the first study demonstrating the effects of peficitinib on RA FLS. We showed that peficitinib suppressed the JAK-STAT pathway of RA FLS and was involved in the suppression of monocyte chemotaxis and proliferation of RA FLS through inhibition of inflammatory cytokines.

First, to determine whether JAK1, JAK2, and JAK3 were expressed in RA ST and FLS, immunohistochemistry was performed. We found that JAK1, JAK2, and JAK3 are expressed in RA STs and FLS. RA FLS plays a crucial role in joint destruction. RA FLS contribute to the perpetuation of disease and play a role in the disease initiation as well [12]. Walker et al. reported that STAT1, STAT4, and JAK3 expression was generally increased in RA STs as compared to the normal STs. Moreover, the STAT1 sublining expression in RA STs was significantly increased as compared to that in the osteoarthritis (OA) STs [13]. Kasperkovitz et al. also reported that the total-STAT1 and phospho-STAT1 expression was predominantly increased in the intimal lining layer and in focal inflammatory infiltrates in RA STs as compared to that in the OA STs. This indicates that the increased expression of STAT1 is intrinsic to RA FLSs in the intimal lining layer, and activation of the pathway by phosphorylation is an active process in RA FLSs [14]. Walker et al. demonstrated that the JAK3, STAT1, STAT4, and STAT6 sublining expression was decreased in response to successful treatment of RA with standard disease-modifying antirheumatic drugs [15]. Based on these reports, the regulation of the JAK-STAT pathway could be a viable therapeutic target for the treatment of RA.

Next, to determine whether peficitinib regulates the JAK-STAT pathway in FLS, western blot was performed. We demonstrated that phosphorylation of STAT1, STAT3, and STAT5 was increased 10 min after stimulation with IL-6 and IL-6R. We showed that the phosphorylation of STAT1, STAT3, and STAT5 in RA FLS was suppressed by peficitinib in a concentration-dependent manner. Interestingly, peficitinib reduced total STAT1 but not STAT3 and STAT5. Previous studies have not reported that JAK inhibitors reduced tSTAT1. This is a possible characteristic of peficitinib. Elucidation of the mechanism by which peficitinib reduced tSTAT is a future subject. Rosengren et al. demonstrated that phosphorylation of STAT in RA FLS is suppressed by other JAK inhibitors [16,17,18]. The concentration of peficitinib in our study was higher compared to the reported IC50 of tofacitinib. Ito et al. demonstrated that the effective concentration of tofacitinib in FLS was 1 μM. In our study, considering the comparison with other JAK inhibbitor, the concentration of peficitinib was set to 0.1, 1 and 5 μM. We decided 5 μM because of Figure 3. Due to the good clinical efficacy of tofacitinib, many JAK inhibitors have been developed in recent years [19]. Tofacitinib inhibits JAK1, JAK2, JAK3, and Tyk2 enzyme activity having IC50 values of 3.7, 3.1, 0.8, and 16 nmol/L, respectively, and has moderate selectivity for JAK3 inhibition similar to peficitinib [6]. Baricitinib also inhibits JAK1, JAK2, JAK3, and Tyk2 enzyme activity having IC50 values of 5.9, 5.7, >400, and 53 nM, respectively [20,21]. Baricitinib is a JAK1 and JAK2 selective inhibitor having a weak JAK 3 inhibitory activity. Baricitinib suppressed STAT3 activation by IL-6 and MCP-1/CCL2 production [22]. The JAK inhibitor tofacitinib inhibits IL-6-induced phosphorylation of STAT1 and STAT3 in a dose-dependent manner [16]. In other autoimmune diseases, tofacitinib was found to inhibit the phosphorylation of STAT1 and STAT3 in psoriatic arthritis FLS as compared to vehicle control [18]. Further, baricitinib, a JAK1/JAK2 inhibitor that has been approved for the treatment of RA [23], inhibited IFNγ-induced phosphorylation of STAT1 in a dose-dependent manner [17]. We found that peficitinib suppressed the phosphorylation of STAT in FLS like tofacitinib, and baricitinib inhibited phosphorylation of STAT in FLS.

We next focused on the functional mechanism of monocyte migration. We showed that peficitinib-treated RA FLS conditioned medium reduced THP-1 and PBMC migration as compared to the untreated RA FLS-conditioned medium. This is the first study to prove that peficitinib suppressed monocyte migration. The process of synovial inflammation results from the influx of a series of inflammatory cells. The synovial fibroblasts of the inflamed RA FLSs release IL-8/CXCL8 and MCP-1/CCL2 and recruit neutrophils, lymphocytes, and mononuclear phagocytes into the joints [24]. Monocyte/macrophage and neutrophil infiltration occur early in the disease progression and correlates with bone and articular erosions. The influx of monocytes and macrophages also highly correlates with inflammation and tissue damage in RA patients [25]. Thus, suppression of monocyte migration by peficitinib is important in the pathogenesis of RA.

In addition, we found that peficitinib suppressed the secretion of inflammatory mediators in RA FLS. MCP-1/CCL2 in RA FLS supernatant was suppressed after treatment with peficitinib as compared to that without treatment. The suppression of chemotaxis of THP-1 and PBMC was also observed through inhibition of MCP-1/CCL2 in the RA FLS supernatant. MCP-1/CCL2 is a cytokine that induces cell migration, such as monocytes and T cells. Koch et al. showed higher MCP-1/CCL2 levels in synovial fluids of RA patients as compared to that of OA patients [24]. MCP-1/CCL2 is produced from the RA FLS, and the production is accelerated by stimulation with TNF-α, IL-1β, etc. [26,27]. MCP-1/CCL2 is thought to be involved in the infiltration of monocytes and macrophages into RA synovium [28]. In our study, inhibition of MCP-1/CCL2 by peficitinib was found to be consistent with the suppression of THP-1 and PBMC migration.

Rosengren et al. showed that tofacitinib inhibited the TNF-induced expression of several chemokines (IP-10/CXCL10, RANTES/CCL5, and MCP-1/CCL2) [16]. TNF did not induce immediate phosphorylation of STAT1 or STAT3. However, TNF activates the JAK / STAT pathway via stimulating the secretion of type I IFN by FLS. Tofacitinib inhibits type I IFN signaling, thereby limiting chemokine induction. We observed the suppression of MCP-1/CCL2 and RANTES/CCL5. These findings suggest that the inhibition of the JAK/STAT pathway by peficitinib and tofacitinib are similar and both are highly selective to JAK3. However peficitinib inhibits Tyk2 more strongly than tofacitinib. It is further research subject to examine how peficitinib acts on FLS stimulated by various cytokines such as TNF. Finally, the peficitinib-treated RA FLS showed a 14 ± 2% reduction in the proliferation of RA FLS as compared to the untreated RA FLS. Previous studies have demonstrated that abnormally proliferating T lymphocytes and FLS are the main causes of RA [29,30]. FLS plays an important role in many pathological processes in RA ST. FLS forms pannus and reduces the ability to undergo apoptosis, production of proteases, and invasion of cartilages [31]. The proliferated and activated FLS show tumor cells-like characteristics, such as decreased apoptosis, secretion of cytokines, MMPs, and proteoglycans, thereby destroying articular cartilage and bone structure [32]. Therefore, inhibition of proliferated and activated FLS may contribute towards better therapeutic effects in RA [33].

## 5. Conclusions

We demonstrated that peficitinib suppressed the JAK-STAT pathway of FLS, and was involved in the suppression of chemotaxis of mononuclear cells and proliferation of FLS through inhibition of inflammatory cytokines in RA.

## Figures and Tables

**Figure 1 cells-08-00561-f001:**
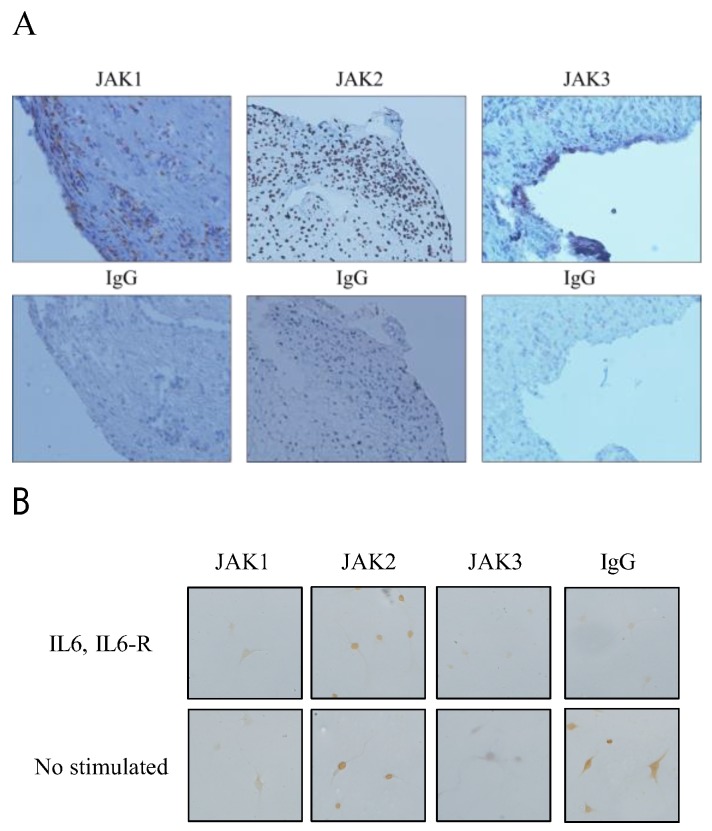
JAK1, JAK2, and JAK3 were expressed in rheumatoid arthritis (RA) synovial tissue (ST) and fibroblast-like synoviocytes (FLS). Frozen sections of RA ST and RA FLS isolated from ST were stained for JAK1, JAK2, or JAK3. (**A**) JAK1, JAK2, and JAK3 were expressed in RA ST. JAK1 and JAK3 were observed in the RA ST lining layers. JAK2 was expressed entirely in the RA ST cell nucleus. (**B**) JAK1, JAK2, and JAK3 were expressed in RA FLS (original magnification 200×).

**Figure 2 cells-08-00561-f002:**
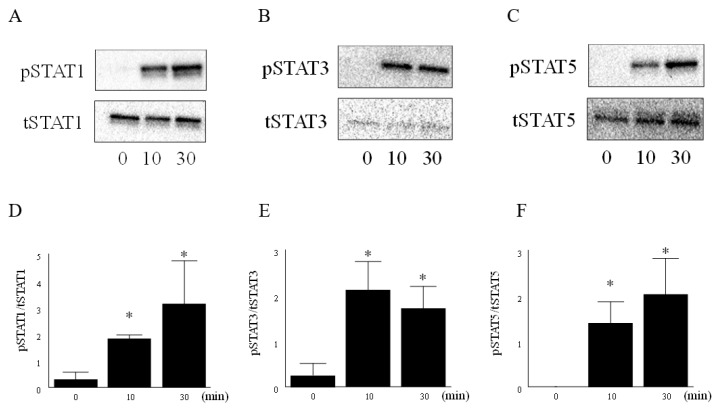
IL-6 and IL-6R activate JAK-STAT pathway in RA FLS. The RA FLS were stimulated with IL-6 (100 ng/mL) and IL-6R (100 ng/mL) for 10 or 30 min. (**A**) Representative western blot showing phospho STAT1 (pSTAT), (**B**) phospho STAT3 (pSTAT3), and (**C**) phospho STAT5 (pSTAT5). (**D**) Expression of pSTAT1 band intensities was quantified and the data are expressed as the mean and SEM. pSTAT1, (**E**) pSTAT3, and (**F**) pSTAT5 were increased 10 min after stimulation with IL-6 and IL-6R. The data are expressed as the mean ± SEM (n = 3 patients). * *p* < 0.05 when unstimulated (0 min).

**Figure 3 cells-08-00561-f003:**
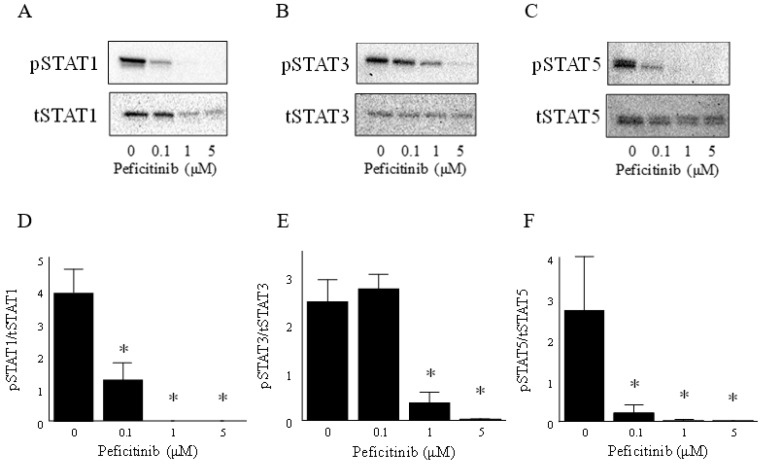
Effects of peficitinib on IL-6 and IL-6R responses in RA FLS. The RA FLS were stimulated with IL-6 (100 ng/mL) and IL-6R (100 ng/mL) after treating with peficitinib (0.1, 1, 5 μM) for 24 h. (**A**) Representative western blot images showed that peficitinib suppressed the phosphorylation of STAT1, (**B**) STAT3, and (**C**) STAT5 in RA FLS. (**D**) The expression of pSTAT1 band intensities was quantified and the data are expressed as the mean and SEM. pSTAT1, (**E**) pSTAT3, and (**F**) pSTAT5 were suppressed by peficitinib (0.1, 1, and 5 µM) in a concentration-dependent manner. The data are expressed as the mean ± SEM (n = 3 patients). * *p* < 0.05 vs. control.

**Figure 4 cells-08-00561-f004:**
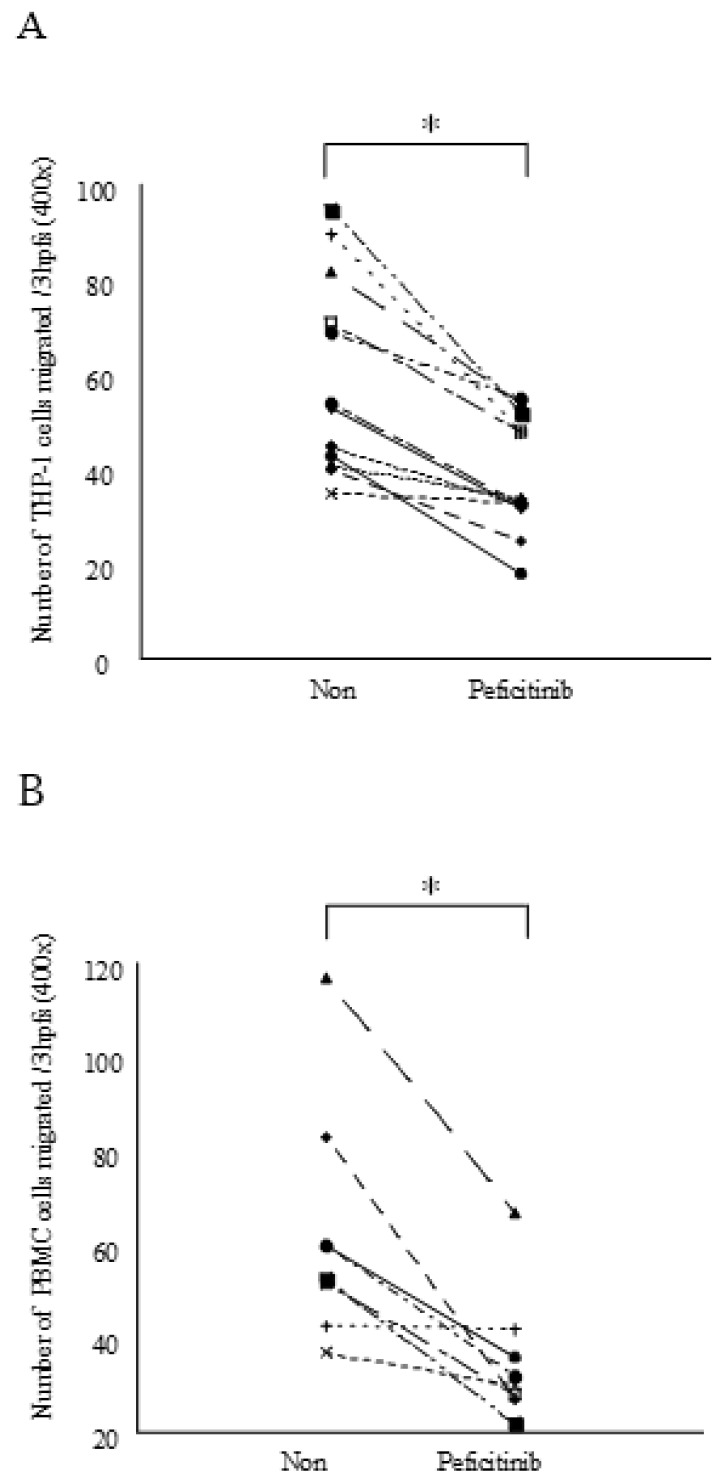
Blocking JAK-STAT signaling with peficitinib in RA FLS reduced monocyte chemotaxis. The RA FLS were stimulated with IL-6 (100 ng/mL) and IL-6R (100 ng/mL) after treating with peficitinib (5 μM) for 24 h. (**A**) The peficitinib-treated RA FLS-conditioned medium reduced THP-1 migration as compared to the untreated RA FLS-conditioned medium (n, number of experiments; 12 patients). (**B**) Peficitinib-treated RA FLS-conditioned medium reduced PBMC migration as compared to the untreated RA FLS-conditioned medium (n, number of experiments; 12 patients). * *p* < 0.05 vs. control.

**Figure 5 cells-08-00561-f005:**
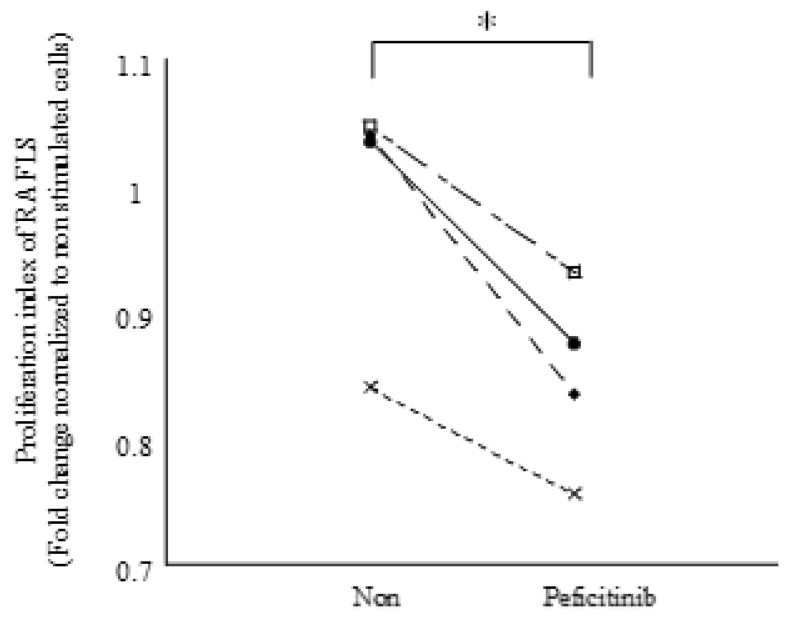
Peficitinib suppressed RA FLS proliferation. RA FLS were stimulated with IL-6 (100 ng/mL) and IL-6R (100 ng/mL) after treating with peficitinib (5 μM) for 24 h. The proliferation of peficitinib-treated RA FLS was reduced as compared to the untreated RA FLS (n, number of experiments; 4 patients). * *p* < 0.05 vs. control.

**Figure 6 cells-08-00561-f006:**
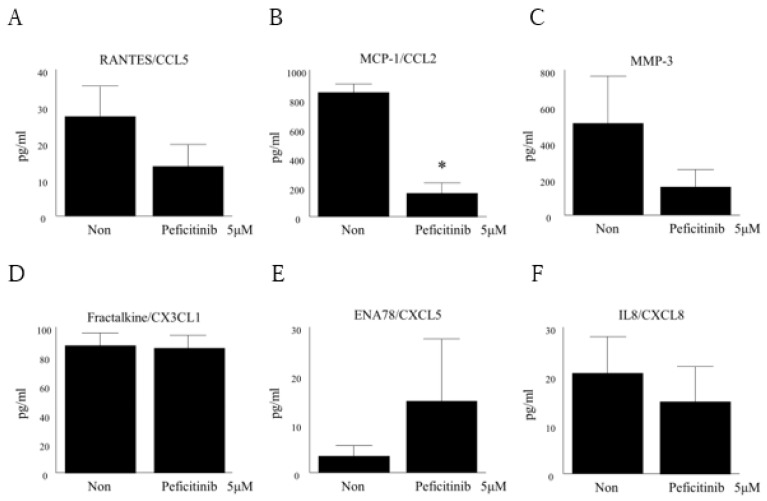
Peficitinib suppressed the secretion of inflammatory mediators in RA FLS. The amounts of RANTES/CCL5 (**A**), MCP-1/CCL2 (**B**), MMP-3 (**C**), fractalkine/CX3CL1 (**D**), ENA78/CXCL5 (**E**), and IL-8 (**F**) in IL-6 and IL-6R-stimulated peficitinib treated RA FLS-conditioned medium were determined and were compared to IL-6 and IL-6R-stimulated untreated RA FLS-conditioned medium. MCP-1/CCL2 in RA FLS supernatant was suppressed after adding peficitinib. (**D**) Fractalkine/CX3CL1 in IL-6 and IL-6R-stimulated peficitinib-treated RA FLS-conditioned medium was not different as compared to the untreated medium. (**E**) ENA-78 in IL-6 and IL-6R-stimulated peficitinib-treated RA FLS conditioned medium was increased. The data are expressed as the mean ± SEM (n = 4 patients). * *p* < 0.05 vs. control.

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
