# Peer review of "Peficitinib Inhibits the Chemotactic Activity of Monocytes via Proinflammatory Cytokine Production in Rheumatoid Arthritis Fibroblast-Like Synoviocytes"

_cells, 2019, doi:10.3390/cells8060561_

Round 1
Reviewer 1 Report
The research reported in this paper was designed to determine the effect of Peficitinib (PEF), a JAK small molecule inhibitor, on several parameters of rheumatoid arthritis fibroblast-like synoviocytes (RA-FLS). A few issues need to be addressed:
1) Page 2/12. The authors should note that both STAT homo-dimers and hetero-dimers can form after STAT proteins are activated.
2) Page 2/12. Please provide more details about the RA patients. For example, what was the duration of disease for patients undergoing arthroplasty versus synovectomy? Were any RA patients receiving biologic drugs, to neutralize TNF-alpha or IL-6 prior to arthroplasty or synovectomy?
3) Page 3/12. Please justify the concentration of PEF (i.e. 0.1-5μM) used in these studies. These concentration are high compared to the reported IC50 of other JAK SMIs. See reference 6.
4) Page 3/12. Statistical analyses. Were equal variances between the groups actually determined and shown to be the same or similar?
5) Page 4/12. Figure 1A. Frozen sections (FS) for control and specific stained FS should have similar morphology. As shown in Figure 1, they do not. Figure 1B. Please elaborate on what IL-6+IL-6R(+) and IL-6+IL-6R(-) mean?
6) Page 5/12. Figure 1A. Two p-STAT1 bands or 1 p-STAT1 band? The same with p-STAT5 (p-STAT5A, p-STAT5B)?
7) Page 6/12. Figure 3 Legend. Why treat with PEF for 24 hrs.? Also does the western blot shown in Figures 3A-3C indicate that PEF also reduced total STAT1 (tSTAT1) but not tSTAT3 or tSTAT5. If so, that would be important data to discuss.
8) Discussion. The authors should discuss any and all disputes in the literature regarding the specificity and selectivity of tofacitinib and PEF.
9) Conclusions. The "inflammatory process" can only be studied in vivo. What this data appears to show is that several parameters associated with inflammation in RA-FLS have been altered by treatment of RA-FLS with PEF.
Author Response
Reviewer 1
1) Page 2/12. The authors should note that both STAT homo-dimers and hetero-dimers can form after STAT proteins are activated.
Thank you for your suggestion.
We have changed that homodimer is homo-dimers and hetero-dimers in Introduction section on page 2.
2) Page 2/12. Please provide more details about the RA patients. For example, what was the duration of disease for patients undergoing arthroplasty versus synovectomy? Were any RA patients receiving biologic drugs, to neutralize TNF-alpha or IL-6 prior to arthroplasty or synovectomy?
We have added patient profiles in materials and methods section on page 2.
One patient was treated with tocilizumab and methotrexate for 11 months and had 2 years disease duration. The other was treated with abatacept for 3 years and had 6 years disease duration. Both patients had total knee arthroplasty.
3) Page 3/12. Please justify the concentration of PEF (i.e. 0.1-5μM) used in these studies. These concentration are high compared to the reported IC50 of other JAK SMIs. See reference 6.
Thank you for your pointing out. The concentration of peficitinib in our study was higher compared to the reported IC50 of tofacitinib. Ito et al. demonstrated that the effective concentration of tofacitinib in FLS was 1 μM. In our study, considering the comparison with other JAK inhibbitor, the concentration of peficitinib was set to 0.1, 1 and 5 μM. We decided 5 μM because of Figure 3. We added this sentence in discussion section on page 11.
4) Page 3/12. Statistical analyses. Were equal variances between the groups actually determined and shown to be the same or similar?
The variance of the data was evaluated using the F test. The data were analyzed using Student’s t-test for equal variance and Mann-Whitney U test for unequal variance. We added this sentence in materials and methods section on page 3.
5) Page 4/12. Figure 1A. Frozen sections (FS) for control and specific stained FS should have similar morphology. As shown in Figure 1, they do not. Figure 1B. Please elaborate on what IL-6+IL-6R(+) and IL-6+IL-6R(-) mean?
We have changed control and specific stained FS to similar morphology in Figure 1A.
IL-6 +IL-6R (+) means that it was stimulated by IL-6 and IL-6R, and IL-6 +IL-6R (-) means that it was not stimulated by IL-6 and IL-6R in Figure 1B. We altered the comment on Figure 1B.
6) Page 5/12. Figure 1A. Two p-STAT1 bands or 1 p-STAT1 band? The same with p-STAT5 (p-STAT5A, p-STAT5B)?
P-STAT1 was two bands in Figure 2. This result was same to the previous study [16]. P-STAT5 had one band (p-STAT5A). Total STAT5 had two bands (p-STAT5A and p-STAT5B). These results were considered to be the influence of the antibody preparation. We also added this information in results section on page 6.
7) Page 6/12. Figure 3 Legend. Why treat with PEF for 24 hrs.? Also does the western blot shown in Figures 3A-3C indicate that PEF also reduced total STAT1 (tSTAT1) but not tSTAT3 or tSTAT5. If so, that would be important data to discuss.
The reason we treated with peficitinib for 24 hours is to keep the cells treated with peficitinib in a steady state.
We agreed with Reviewer 1. Further we confirmed, peficitinib also reduced total STAT1 (tSTAT1) but not tSTAT3 or tSTAT5. Previous studies have not reported that JAK inhibitors reduced tSTAT1. This is a possible characteristic of peficitinib. Elucidation of the mechanism by which peficitinib reduced tSTAT is a future subject. We added this sentence in discussion section on page 11.
8) Discussion. The authors should discuss any and all disputes in the literature regarding the specificity and selectivity of tofacitinib and PEF.
Thank you for your suggestion.
We added the following sentences referring to the past study of tofacitinib in discussion section on page 12.
TNF did not induce immediate phosphorylation of STAT1 or STAT3. However, TNF activates the JAK / STAT pathway via stimulating the secretion of type I IFN by FLS. Tofacitinib inhibits type I IFN signaling, thereby limiting chemokine induction.
However peficitinib inhibits Tyk2 more strongly than tofacitinib.
It is further research subject to examine how peficitinib acts on FLS stimulated by various cytokines such as TNF.
9) Conclusions. The "inflammatory process" can only be studied in vivo. What this data appears to show is that several parameters associated with inflammation in RA-FLS have been altered by treatment of RA-FLS with PEF.
We removed the last sentence of the conclusions on page 12.
Reviewer 2 Report
Comments to the Author
I thank the editor for the chance of reviewing this manuscript showing the ability of Peficitinib in suppressing the JAK-STAT pathway in RA FLS as well as monocyte chemotaxis and proliferation of FLS through inhibition of inflammatory cytokines. This is the first study demonstrating the effects of peficitinib on RA FLS. In my opinion this is a very interesting study worthy of being considered for publication. I only have some comments that should be easy to address, listed in the order of appearance below.
- Many cytokines that bind type I and type II cytokine receptors are critical regulators of RA and employ the Janus kinase and signal transducer and activator of transcription (STAT) pathway to exert their effect. Authors should explain these mechanisms in their introduction.
- Authors should state if patients was treated before undergoing synovial tissues collection, as immunosuppressive treatments could have affected cytokines pathways.
- Minor English changes and spell check are required
Author Response
Reviewer 2
1) Many cytokines that bind type I and type II cytokine receptors are critical regulators of RA and employ the Janus kinase and signal transducer and activator of transcription (STAT) pathway to exert their effect. Authors should explain these mechanisms in their introduction.
We have explained type I and type II cytokine receptors critical regulators of RA in introduction section on page 2. Pharmacological inhibition of JAKs was shown to efficiently block the downstream events associated with type I/II cytokines.
2) Authors should state if patients was treated before undergoing synovial tissues collection, as immunosuppressive treatments could have affected cytokines pathways.
We have added patient profiles in materials and methods section on page 2. We have inserted the following sentence.
One patient was treated with tocilizumab and methotrexate for 11 months and had 2 years disease duration. The other was treated with abatacept for 3 years and had 6 years disease duration. These treatments could have affected cytokines pathways.
3) Minor English changes and spell check are required
We already English review by native speaker, however, we did not have time to do it again. English review will be done immediately.
Round 2
Reviewer 1 Report
No further comments